# Enhancement of the Efficacy of Photodynamic Therapy against Uropathogenic Gram-Negative Bacteria Species

Vadim Elagin [1,*], Ivan Budruev [2], Artem Antonyan [3], Pavel Bureev [2], Nadezhda Ignatova [4], Olga Streltsova [3] and Vladislav Kamensky [1,5]

1 Institute of Experimental Oncology and Biomedical Technologies, Privolzhsky Research Medical University, Nizhny Novgorod 603005, Russia
2 Institute of Biology and Biomedicine, National Research Lobachevsky State University of Nizhny Novgorod, Nizhny Novgorod 603005, Russia
3 Department of Urology Named after E. V. Shakhov, Privolzhsky Research Medical University, Nizhny Novgorod 603005, Russia
4 Department of Epidemiology, Microbiology and Evidence-Based Medicine, Privolzhsky Research Medical University, Nizhny Novgorod 603005, Russia
5 Federal Research Center Institute of Applied Physics of the Russian Academy of Sciences, Nizhny Novgorod 603950, Russia
* Correspondence: elagin.vadim@gmail.com; Tel.: +7-(831)-465-56-72

**Abstract:** Antimicrobial photodynamic therapy (aPDT) was demonstrated to be effective against various species of Gram-positive bacteria. However, the complex structure of a Gram-negative bacteria envelope limits the application of aPDT. Thus, the goal of this study was to improve the efficiency of antimicrobial photodynamic therapy with Fotoditazin against uropathogenic Gram-negative bacteria. The non-ionic detergent Triton X-100 and emulsifier Tween 80 were tested. The effect of extracellular photosensitizer on aPDT efficacy was analyzed. Moreover, the irradiation regime was optimized in terms of the output power and emitting mode. It was found that Triton X-100 at 10% vol enhanced the efficacy of aPDT of *E. coli* up to 52%. The subsequent observation demonstrated that, when the photosensitizer was removed from the extracellular space, the efficacy of aPDT on various Gram-negative species decreased dramatically. As for the irradiation mode, an increase in the laser output power led to an increase in the aPDT efficacy. The pulsed irradiation mode did not affect the aPDT efficacy. Thus, in order to achieve optimal aPDT efficacy, bacteria should be irradiated at 450-mW output power in the presence of Triton X-100 and a photosensitizer in the extracellular environment. However, it should be noted that the efficacy of aPDT of *K. pneumoniae* was significantly lower than for other species. The developed aPDT technique may be effective in a native environment of uropathogenic microorganisms.

**Keywords:** gram-negative bacteria; antimicrobial photodynamic therapy; fotoditazin; laser; triton X-100; Tween 80

## 1. Introduction

The growing resistance of pathogenic microorganisms to the commonly used antibiotics [1,2] has stimulated the development of alternative approaches that exhibit high antimicrobial activity. One such alternative approach is antimicrobial photodynamic therapy (aPDT) [3,4]. aPDT, similarly to a common anticancer PDT, involves an interaction between a photosensitizer and light of an appropriate wavelength which produce singlet oxygen [5]. In contrast to antibiotics, aPDT has various targets in bacterial cells, such as membranes, enzymes, lipids, and DNA [6,7]. The multi-target mechanism of aPDT reduces the risk of developing a resistance against microorganisms exposed to it [8]. It is believed that aPDT is more effective if a photosensitizer is taken up into a bacterial cell before light is delivered. In the previous papers, it was shown that PDT may be used

for the elimination of multidrug-resistant bacteria, including uropathogenic strains [9,10]. However, this approach is sometimes ineffective, especially against Gram-negative bacteria due to the complex structure of their envelope. Gram-positive bacteria have a relatively thicker but porous cell wall, made up of interconnected peptidoglycan layers surrounding a cytoplasmic membrane. The teichoic acid residues of the cell wall provide a negative charge and form binding sites for cationic molecules [11]. The cell envelope of Gram-negative bacteria is composed of an outer membrane, a thinner peptidoglycan layer and a cytoplasmic membrane. The transport of molecules across the cell wall of Gram-negative bacteria is regulated at the outer membrane, which is rich in lipopolysaccharides [12]. Unlike Gram-positive bacteria, the membrane barrier of Gram-negative bacteria prevents the uptake of anionic and neutral photosensitizers. It is well known that urinary tract infections are primarily caused by Gram-negative bacteria *Escherichia coli*, *Klebsiella pneumoniae*, *Proteus mirabilis*, and *Pseudomonas aeruginosa* [13]. For example, urinary calculi may be induced by a urinary tract infection or contaminated by bacteria. In both cases, lithotripsy leads to the spread of bacteria in organ cavities and postoperative complications such as pyelonephritis, systemic inflammatory reaction syndrome, and urosepsis [14]. The wide spread of multidrug-resistant bacteria makes antibiotic therapy inefficient. Therefore, aPDT should be optimized for the elimination of Gram-negative uropathogenic bacteria. One of the ways to overcome this difficulty is either to use cationic photosensitizers, or to bind the photosensitizer with positively charged entities. It was shown that the cationic photosensitizer, methylene blue, uptakes better than anionic photosensitizers, Rose Bengal and indocyanine green by Gram-positive, *Enterococcus faecalis*, and provides complete inactivation of bacteria by aPDT [15]. Another way is to use divalent cations in extracellular media. In an earlier paper, it was demonstrated that divalent cations increase the uptake of anionic photosensitizers by Gram-positive bacterial cells [15]. Moreover, various surfactants may be used to enhance the interaction between a photosensitizer and bacterial cells. It is known that the oleic acid moiety of Tween 80 can incorporate into the cell membrane, which affects cell membrane properties [16]. The deformation and membrane rupture of bacteria treated by the nanoemulsion of eugenol and Tween 80 was shown [17]. In addition, the permeability of the eukaryotic cell membrane to the highly charged hydrophilic molecule may be stimulated by Triton X-100 [18]. Triton X-100 is able to incorporate the liposomes of cell membranes, increasing the penetrability of the membrane [19]. Despite this, Triton X-100 shows weak antibacterial activity and is seldom used as an antibacterial agent [20].

Fotoditazin used in this study as a photosensitizer is N-dimethylglucamine salt of chlorin e6, approved for clinical application by the Ministry of Health of the Russian Federation. It is soluble in water and is able to penetrate in the biological membrane, which is expected to improve PDT action. Fotoditazin possesses an intensive absorption band in the long-wave red field of the spectrum [21]. This photosensitizer has shown promising results in the inactivation of *Candida albicans* [22–24] as well as multidrug-resistant *Staphylococcus aureus* [4,25].

It is well known that aPDT is more effective if sufficient concentrations of molecular oxygen is present near photosensitizer molecules. High-powered light irradiation in continuous mode results in the intense consumption of molecular oxygen and, hence, hypoxy, which makes aPDT inefficient. Moreover, continuous mode aPDT leads to overheating and photosensitizer bleaching. In the pulsed irradiation mode, both molecular and reactive oxygen can diffuse from the neighboring regions, thus making the photosensitizer more active [26]. Optimal efficiency of the pulsed mode irradiation in comparison with the continuous one was demonstrated on cancer cells [27–29].

The purpose of this study was to improve the efficiency of antimicrobial photodynamic therapy against uropathogenic Gram-negative bacteria by means of surfactants, as well as variation of laser irradiation parameters.

## 2. Materials and Methods

### 2.1. Bacterial Strains and Culture Conditions

The strains of *Escherichia coli*, *Klebsiella pneumoniae*, *Pseudomonas aeruginosa* and *Proteus mirabilis* were obtained from the collection of clinical uropathogenic bacterial isolates of the Research Institute of Experimental Oncology and Biomedical Technologies, Privolzhsky Research Medical University. The strain was maintained on agar plates at 4 °C. For experiments, the bacteria were grown overnight at 37 °C and 150 rpm on nutrient broth. Next, the bacteria were harvested by centrifugation (7000 rpm/7 min) and re-suspended in phosphate-buffered saline to reach 0.5 McFarland (corresponding to a concentration of $1$–$2 \times 10^8$ CFU/mL). aPDT treated and control samples were inoculated onto nutrient agar and kept at 37 °C for 17–20 h to count the number of colony forming units. Finally, the efficiency of the selected aPDT regime was tested on the patients' urine culture, which tested positively on the studied bacterial species.

### 2.2. Sample Preparation

A photosensitizer (Fotoditazin®, LLC Veta-Grand, Moscow, Russia) was added to the bacterial suspensions to a final concentration of 50 µg/mL and kept for 10 min in the dark at room temperature. The non-ionic detergent Triton X-100 and emulsifier Tween 80 were used as agents, increasing the accumulation of the photosensitizer inside bacterial cells. These substances were added to the samples at various final concentrations (1%, 5% and 10%). The effect of the photosensitizer in the extracellular medium on the aPDT efficacy was also evaluated. For this, some of the samples were washed from the unbound photosensitizer before irradiation by centrifugation, and subsequent resuspension in a fresh medium. The bacterial suspension were incubated with Fotoditazin (50 µg/mL) and Triton X-100 (10% vol) for 15 or 30 min in the dark at room temperature. The twice-washed by PBS solution microorganisms were resuspended in 500 µL of PBS solution for fluorescence assessment. The photosensitizer accumulation was analyzed by the fluorescence intensity change using the IVIS Spectrum (Caliper Life Sciences, Waltham, MA, USA) with excitation at 640/35 nm and emission at 680/20 nm. The average intensity of fluorescence was calculated for the same size areas of each sample using Living Image software (Caliper Life Sciences, USA). Calibration curves were constructed for photosensitizer in PBS. After fluorescence measurement, bacterial cells were inoculated on agar to calculate the number of CFU. The amount of Fotoditazin taken up by a single bacterial cell were calculated as the total amount of the photosensitizer in the sample divided by the number of CFU.

### 2.3. Antimicrobial Photodynamic Therapy

aPDT was performed using a medical laser device (Latus, LLC Aktus, Russia) at a wavelength of 662 nm. For irradiation, 100 µL of each sample were placed into the wells of 96-well culture plates with black walls. Irradiation was performed using an optical fiber equipped with a collimator and placed above the plate. Each well was illuminated separately by a continuous wave laser at an output power of 150–450 mW for 10 min. To evaluate the effect of pulsed irradiation on aPDT efficacy, a pulse width of 100 ms and a repetition rate of 5 Hz were selected. Irradiation was carried out at an output power of 300 mW and 450 mW, until light doses equal to continuous wave irradiation regimes were achieved. The efficacy of aPDT was calculated according to Equation (1). The correspondence of aPDT efficacy values to the quantitative reduction of *CFU* is clear from Table 1:

$$Efficacy = \left(1 - \frac{CFU_{treat}}{CFU_{ctrl}}\right) \times 100, \tag{1}$$

where $CFU_{treat}$ is the number of bacterial colonies after aPDT, $CFU_{ctrl}$ is the number of bacterial colonies in the control group.

**Table 1.** Bacteria reduction equivalency chart.

| aPDT Efficacy | Log Reduction |
|:---:|:---:|
| 0 | 0 |
| 90 | 1 |
| 99 | 2 |
| 99.9 | 3 |
| 99.99 | 4 |
| 99.999 | 5 |

*2.4. Statistical Analysis*

The aPDT efficacy values were presented as a mean ± standard deviation. To calculate the statistical significance of the differences, the ANOVA with Bonferroni post hoc test was used. Statistical analysis was performed with Statistica 10 (StatSoft. Inc., Tusla, OK, USA). $p$-values ≤ 0.05 were considered statistically significant.

**3. Results**

*3.1. The Effect of Non-Ionic Detergent and Emulsifier*

The efficacy of aPDT of *E. coli* washed from an unbound photosensitizer under continuous wave irradiation was only 5%. The addition of either Tween 80 or Triton X-100 to bacteria without the photosensitizer did not cause a significant variation in the number of living cells. Moreover, low concentrations (1% and 5%) of the surfactants, coupled with the photosensitizer, did not induce bacteria killing after aPDT. However, the presence of these agents at 10% in the samples during photosensitizer incubation resulted in a decreased number of live bacteria after aPDT. Tween 80 at a final concentration of 10% vol provided an insignificant enhancement of aPDT efficacy of up to 9% (Figure 1). The addition of Triton X-100 significantly decreased the amount of live bacteria. The efficacy of aPDT of *E. coli* incubated with the photosensitizer and Triton X-100 achieved 52.5%. Thus, Triton X-100 at 10% vol may be used to improve aPDT efficacy.

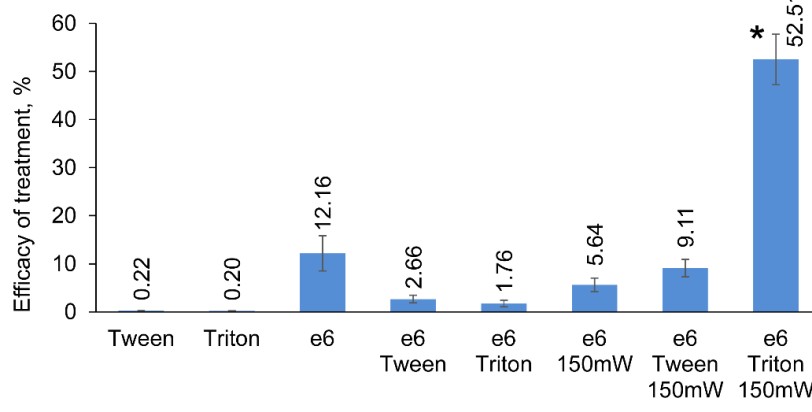

**Figure 1.** The influence of various components of aPDT and their combinations on *E. coli* viability. The following concentrations of agents were used for incubation with bacteria, 10% of Tween 80, 10% of Triton X-100 and 50 µg/mL of Fotoditazin (e6). Bacteria were incubated for 15 min at room temperature in the dark. Before irradiation by continuous wave laser at 150 mW, bacteria were washed from unbound chemicals. Statistically significant difference comparable with e6 150 mW (*) is marked.

*3.2. The Effect of the Extracellular Photosensitizer*

The influence of the extracellular photosensitizer during light irradiation was studied on *Escherichia coli*, *Klebsiella pneumoniae*, *Pseudomonas aeruginosa* and *Proteus mirabilis*. The

bacteria were incubated with the photosensitizer and Triton X-100 for 15 min. Before irradiation, a fraction of each sample was washed from the extracellular photosensitizer. It was found that washing the extracellular photosensitizer led to loss of the aPDT efficacy (Figure 2). *K. pneumoniae* was not sensitive to aPDT without the extracellular photosensitizer, while the efficacy of aPDT with the photosensitizer was 89%. *E. coli* had low sensitivity to aPDT without the extracellular photosensitizer as well as with it. The high sensitivity of *P. aeruginosa* to aPDT with the extracellular photosensitizer significantly reduced after washing the photosensitizer. It was revealed that only the efficacy of aPDT of *P. mirabilis* did not change after washing the extracellular photosensitizer.

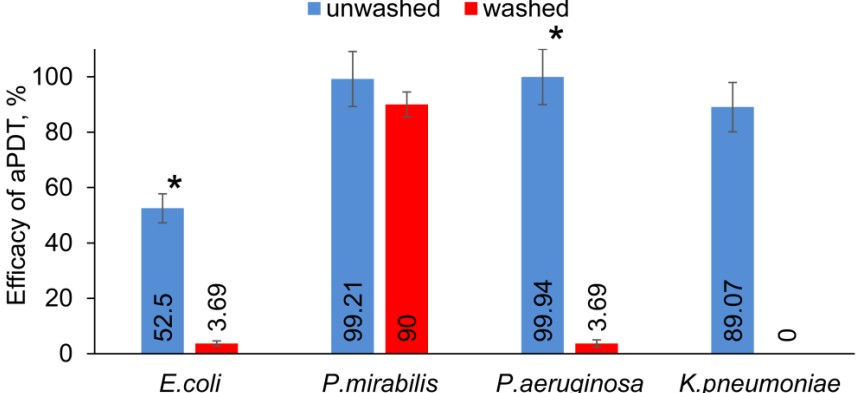

**Figure 2.** The influence of the extracellular photosensitizer on the efficacy of aPDT of Gram-negative bacteria. Bacteria were incubated with Fotoditazin and Triton X-100 for 15 min at room temperature in the dark. Before irradiation by continuous wave laser at 150 mW, part of each sample was washed from unbound chemicals. Statistically significant differences compared to the washed samples (*) are marked.

Analysis of the intracellular concentration of the photosensitizer was carried out on strains of *E. coli*, *K. pneumoniae*, *P. aeruginosa* and *P. mirabilis*. Bacteria were incubated with 50 µg of Fotoditazin for either 15 or 30 min in the dark at room temperature. In addition, Triton X-100 at 10% vol was added to some of the samples. It was found that Triton X-100 increased the intracellular concentration of the photosensitizer over time (Figure 3). At the same time, Triton X-100 free samples demonstrated a decrease in the intracellular concentration of the photosensitizer after 30 min. It should be noted that the intracellular concentration of Fotoditazin after 15 min incubation of the species treated with Triton X-100 differed by less than 3.5 times.

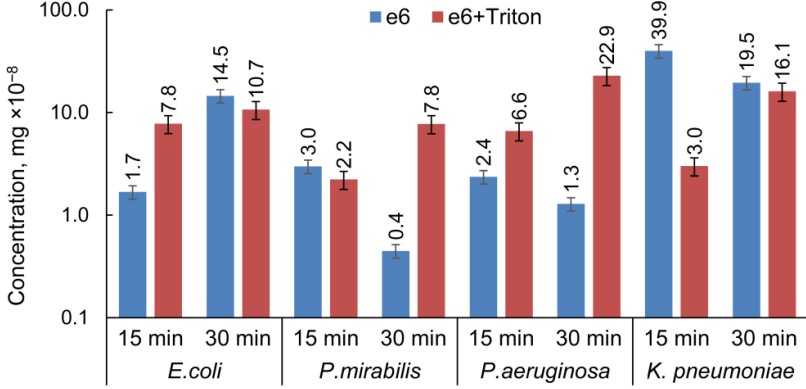

**Figure 3.** The amount of Fotoditazin taken up by a single bacterial cell of Gram-negative species depending on incubation time and presence of Triton x-100. Bacteria were incubated with photosensitizer (e6) and Triton X-100 for 15 min or 30 min at room temperature in the dark. Before measurement the samples were washed from unbound chemicals.

### 3.3. The Effect of Output Laser Power

The bacteria were irradiated at different laser powers after 15 min of incubation with the photosensitizer and Triton X-100, without washing the extracellular photosensitizer. The efficacy of the laser treatment alone against *K. pneumoniae* and *P. aeruginosa* was less than 10% (Figure S1). In contrast, *E. coli* irradiated by laser without Fotoditazin demonstrated an insignificant increase in the number of CFU. The efficacy of the laser treatment alone against *P. mirabilis* was achieved at 20%. It was demonstrated that the aPDT efficacy depended on laser power in all studied species (Figure 4). It was found that the viability of *K. pneumoniae* did not depend on irradiation power. The efficacy of *K. pneumoniae* treatment did not exceed 93%. The irradiation of other bacteria species with a power of 450 mW provided an aPDT efficacy of 99.99%.

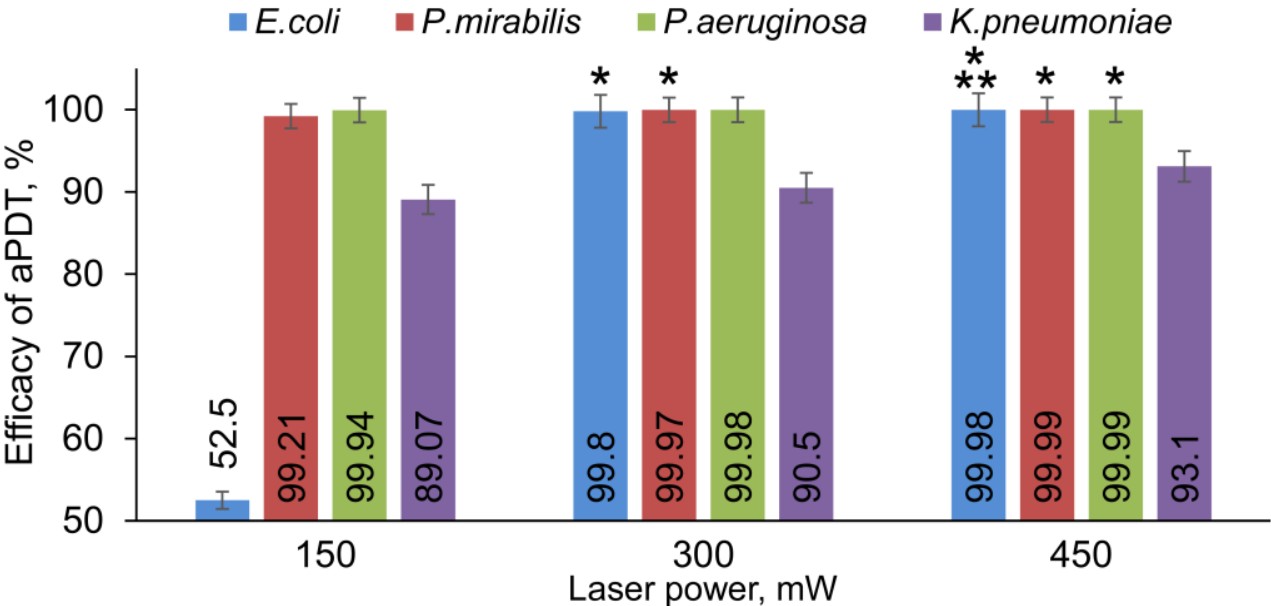

**Figure 4.** The influence of output laser power on the efficacy of aPDT of Gram-negative bacteria. Bacteria were incubated with Fotoditazin and Triton X-100 for 15 min at room temperature in the dark, and illuminated by continuous wave laser at different output powers. Statistically significant differences compared to the corresponding species irradiated by 150 mW (*) and 300 mW (***) are marked.

### 3.4. The Effect of Pulsed Laser Irradiation

For comparison of the aPDT efficacy under the pulsed and continuous mode of irradiation, we chose the output power of 300 mW and 450 mW. The bacteria were incubated with a photosensitizer and Triton X-100, and were not washed after that. *E. coli* and *P. mirabilis* were irradiated by a pulsed laser that had the values of output power and power density similar to the continuous mode (Figure 5). It was found that, at 300 mW, the efficacy of aPDT against *E. coli* was comparable for the pulsed and continuous mode irradiation. The increase of the pulsed laser output power did not affect the aPDT efficacy against *E.coli*. However, it was reduced in comparison with the continuous mode irradiation at 450 mW. The irradiation of *P. mirabilis* by a pulsed laser at 300 mW output power led to a reduced aPDT efficacy, compared to the continuous mode irradiation at the same power. The aPDT efficacy after irradiation at 450 mW by pulsed laser increased in comparison with irradiation at 300 mW. However, it was lower than after the continuous mode irradiation at the same power.

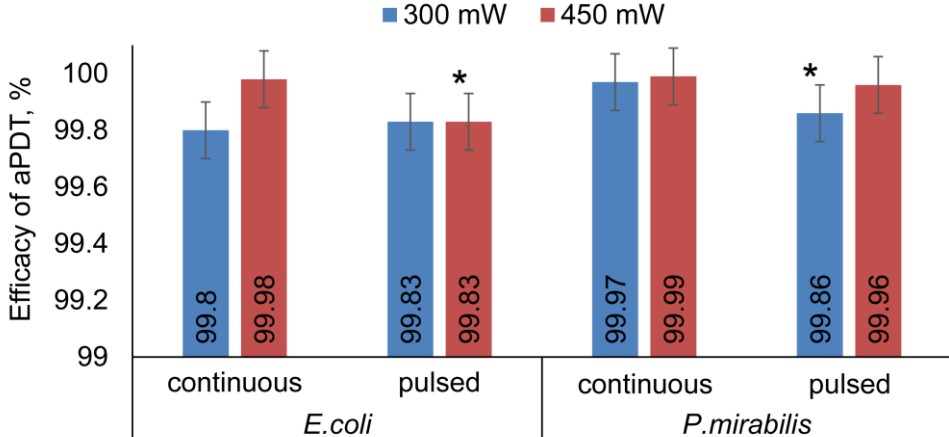

**Figure 5.** The influence of pulsed laser irradiation on the efficacy of aPDT of Gram-negative bacteria. Bacteria were incubated with Fotoditazin and Triton X-100 for 15 min at room temperature in the dark, and illuminated by pulsed laser (100 ms pulse width and 5 Hz repetition rate) at different output powers. Statistically significant differences compared to the corresponding species irradiated by continuous wave laser (*) are marked.

*3.5. The Efficacy of aPDT in Urine Culture*

To test the efficacy of the developed aPDT technique, urine cultures of the patients were incubated with a photosensitizer and Triton X-100 for 15 min in the dark. Then, the unwashed samples were illuminated by a continuous wave laser at 450 mW of output power. Figure 6 demonstrates that the efficacy of the aPDT of infected urine cultures was not less than 99.996%. Thus, the number of microorganisms in the samples was reduced by 10,000–100,000 times.

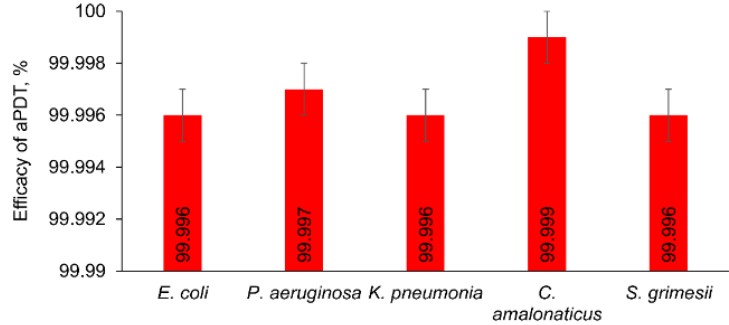

**Figure 6.** Efficacy of aPDT of Gram-negative bacteria in urine culture. Bacteria were incubated with Fotoditazin and Triton X-100 for 15 min at room temperature in the dark, and illuminated by continuous wave laser at 450 mW of output power.

**4. Discussion**

Our previous study showed the efficacy of aPDT against Gram-positive uropathogenic bacteria, while optimization was required for Gram-negative ones [4]. In this work, the conditions for elimination of microorganisms such as *E. coli*, *K. pneumoniae*, *P. aeruginosa*, and *P. mirabilis* were provided. On the one hand, the impact of non-ionic detergent/emulsifier as well as extracellular photosensitizer on aPDT efficacy was studied. On the other hand, the irradiation regime was optimized. It is well known that the crucial factor of aPDT is reactive oxygen species, which have a short half-life [30]. Therefore, the photosensitizer molecules should be located close to the target molecules of bacterial cells. Most researches are devoted to the enhancement of the uptake of the photosensitizer by bacterial cells. Either positively charged photosensitizers [31,32] or neutralization of the cell wall charge [33,34] may be useful in this case. In the present study, Triton X-100 and Tween 80 were used for the

first time as agents that increase the uptake of the photosensitizer into bacterial cells. The Triton X-100 monomer penetrates into the membrane at low concentrations [35], and may lead to the uptake of the photosensitizer. It was shown that irreversible permeabilization of the membrane of eukaryotic cells was achieved at concentrations of Triton X-100 above the critical micelle concentration [18]. The concentration of Triton X-100 used in this study was lower than the critical micelle concentration. Moreover, this concentration of Triton X-100 did not reduce the viability of bacteria. It is known that the structure of Tween 80 has the oleic acid moiety that can be incorporated into the cell membrane and can affect its properties [36]. The used concentration of Tween 80 was below the critical micelle concentration and formed a true solution. As evident from this study, an increase in the efficacy of aPDT in the presence of non-ionic detergents could be achieved by using the 10% concentration of Triton X-100 in the sensitizing solution.

At the next stage of our research, it was revealed that the washing of bacteria from an unbound photosensitizer dramatically reduced the efficacy of aPDT. This implies that the photosensitizer in a solution plays a significant part in the light-mediated killing of the bacteria. There are three possible explanations for this observation. First, it may be due to some damage of the cell wall/membrane induced by extracellular-generated reactive oxygen species, which directly leads to cell death or enhances the photosensitizer penetration into cells, and therefore increases the aPDT efficacy. The second explanation is that the extracellular part of the photosensitizer may continue to diffuse into the cells, while the efflux pumps try to remove it from the cells. Third, long-lived reactive oxygen species generated in extracellular space may diffuse into the cell, resulting in damage-induced cell death. It is likely that all mechanisms contribute to the efficacy of aPDT. The effectiveness of *E. coli* killing was shown earlier using Rose Bengal covalently bound to polystyrene beads that do not penetrate into cells [37]. Moreover, the treatment of bacteria with verapamil, an inhibitor of calcium channels and some efflux pumps, results in increased photosensitizer accumulation [38].

The efficacy of PDT and aPDT is known to be dependent on the irradiation dose [39,40]. However, light exposure cannot be increased during lithotripsy due to time limitation. For this reason, different values of laser power were tested for aPDT efficiency. The minimum used power of 150 mW was ineffective against *E. coli* and *K. pneumoniae*. This may be due to the size and composition of the cell wall of these species. The outer membrane of *E. coli* is filled with lipopolysaccharides, composed of a long polysaccharide chain connected to a complex lipid with several fatty acid tails [41]. A polysaccharide capsule produced by *K. pneumoniae* protects bacteria from unfavorable environmental conditions. The capsule and other bacterial surface polysaccharides also function as a physical barrier to prevent or limit penetration into the cell [42]. It is also possible that the concentration of reactive oxygen species produced by irradiation at 150 mW may be insufficient to induce damage to a large number of cells. Another important factor is the high optical density of bacterial suspension with a photosensitizer. The penetration of light tends to decrease due to optical absorption [43]. The maximal efficacy of aPDT was observed after irradiation at 450 mW. However, the light at this wavelength and power may induce heating of the surrounding tissues. To prevent this, and to provide the possibility of diffusion of oxygen molecules and reactive oxygen species, pulsed irradiation was studied. The earlier theoretical and experimental studies demonstrated the increased efficiency of singlet oxygen generation in a pulsed irradiation mode, compared to a continuous wave mode with the same power density [27]. However, pulsed irradiation in the photodynamic treatment of skin diseases was shown to be less effective than conventional irradiation [44]. Our study demonstrated that the efficacy of aPDT with pulsed irradiation was 0.1% lower, which corresponds to an increase in the number of viable bacteria by 10 times. The repetition rate and pulse width are two important parameters determining the effectiveness of aPDT when pulsed lasers are utilized [45]. The intensity and duration of reactive oxygen species production varied with pulse width, which may lead to a different cytotoxic effect in aPDT [46].

## 5. Conclusions

The impact of non-ionic detergent/emulsifier and the photosensitizer in the extracellular space, as well as the irradiation regimes on aPDT efficacy, were investigated. The presence of Triton X-100, as well as a photosensitizer in the extracellular environment, significantly increased the aPDT efficacy. The output laser power was in direct proportion to the number of killing bacterial cells. A slight decrease of aPDT efficacy was observed during pulsed irradiation. Maximum aPDT efficacy was found under the continuous wave irradiation at 450 mW in the presence of Triton X-100 and the photosensitizer in the extracellular environment. The developed aPDT demonstrated efficacy in the native environment of uropathogenic microorganisms.

**Supplementary Materials:** The following supporting information can be downloaded at: https://www.mdpi.com/article/10.3390/photonics10030310/s1, Figure S1: The influence of output laser power and irradiation mode on the efficacy of aPDT of Gram-negative bacteria without Fotoditazin and Triton X-100.

**Author Contributions:** Conceptualization, V.E. and N.I.; methodology, O.S., V.K. and A.A.; investigation, P.B., I.B., A.A. and V.E.; writing—original draft preparation, V.E. and N.I.; writing—review and editing, V.K.; project administration, O.S.; funding acquisition, V.K. All authors have read and agreed to the published version of the manuscript.

**Funding:** This research was funded by the Russian Science Foundation under grant No. 21-15-00371.

**Institutional Review Board Statement:** This study was approved by the Institutional Review Board of the Privolzhsky Research Medical University (Protocol #13 from 7 July 2021).

**Informed Consent Statement:** Informed consent was obtained from all subjects involved in the study. Written informed consent has been obtained from the patients to publish data in scientific literature.

**Data Availability Statement:** The data presented in this study are available on request from the corresponding author.

**Conflicts of Interest:** The authors declare no conflict of interest.

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
