# Peer review of "Enhancement of the Efficacy of Photodynamic Therapy against Uropathogenic Gram-Negative Bacteria Species"

_photonics, doi:10.3390/photonics10030310_

Round 1

Reviewer 1 Report

This manuscript is very well written, and the presentation is clear. There are a couple of typos and minor corrections that are suggested see below

Materials and Methods

  Line 119 – the 2nd CFUtreat should be changed to ???????

·  Line 122-123 : Table 1. Bacteria reduction equivalency chart - needs to be explained further. What does “Times smaller” mean? 

      Results

·         Lines 203-204: “Thus, the number of microorganisms in the samples was reduced by 10.000-100.000 times” – consider modification of the statement

Author Response

Dear Reviewer,

First of all, we would like to thank You for the very helpful comments that allowed us to improve the quality of the manuscript. We hope that these changes now make the manuscript suitable for publication in Photonics. Below, please, find our detailed reply to the comments. Our respones are given in yellow.

1. Line 119 – the 2nd CFUtreat should be changed to ???????

The 2nd CFUtreat has been changed to ???????

2. Line 122-123 : Table 1. Bacteria reduction equivalency chart - needs to be explained further. What does “Times smaller” mean? 

“Times smaller” is means the decrease of CFU value. We have excluded this column as it is similar to the "log reduction" column.

3. Lines 203-204: “Thus, the number of microorganisms in the samples was reduced by 10.000-100.000 times” – consider modification of the statement

That was a typo. The correct version is: The number of microorganisms in the samples was reduced by 10000-100000 times.

Reviewer 2 Report

The article by Elagin et al. it is not excessively innovative, in fact the use of surfactants to increase the photodynamic effect is known and has been reported in various publications. Anyway, the application, presented by the authors, is original and therefore the article deserves to be published only after some improvements.

1) the purpose of the work, the final part of the introduction, should be developed a little;

2) in the materials and methods table 1 is superfluous;

3) results 3.1: why was only one concentration used for the surfactants? It would be advisable to repeat some tests by modifying the concentrations, perhaps using a mix of Tritan and Tween;

4) results 3.2: in my opinion it would be necessary to try even longer incubation times (30, 45, 60 minutes) to then verify the effects of the bound PS;

5) results 3.4: the sentence in line 187-188 is not consistent with what is reported in figure 4, in fact, the significance in E. coli is reported. Regarding the mirabilis results, some additional comments would be needed;

6) Discussion: in relation to what is written in line 260-262, the production of ROS should be determined.

Author Response

Dear Reviewer,

First of all, we would like to thank You for the very helpful comments that allowed us to improve the quality of the manuscript. We hope that these changes now make the manuscript suitable for publication in Photonics. Below, please, find our detailed reply to the comments. Our responses are given in yellow.

1) the purpose of the work, the final part of the introduction, should be developed a little;

The introduction has been enhanced.

2) in the materials and methods table 1 is superfluous;

We have excluded the “Times smaller” column as it is similar to the "log reduction" column.

3) results 3.1: why was only one concentration used for the surfactants? It would be advisable to repeat some tests by modifying the concentrations, perhaps using a mix of Tritan and Tween;

Three concentrations (1%, 5%, and 10%) of both surfactants were studied. The concentrations less than 10% did not provoke any effects. The corresponding information was added to the manuscript. We did not study concentrations above 10% in view of their possible cell toxicity.

4) results 3.2: in my opinion it would be necessary to try even longer incubation times (30, 45, 60 minutes) to then verify the effects of the bound PS;

In our previous work, the accumulation of the photosensitizer by bacterial cells was investigated. It was found that E. coli and P. mirabilis had maximal accumulation of photosensitizer after 30 min which significantly decreased after 60 min. However, we are planning to use the developed aPDT technique during lithotripsy, therefore, the manipulation time is limited. For this reason, we used short accumulation time.

5) results 3.4: the sentence in line 187-188 is not consistent with what is reported in figure 4, in fact, the significance in E. coli is reported. Regarding the mirabilis results, some additional comments would be needed;

The paragraph was rewritten. The increase of the pulsed laser output power did not affect the aPDT efficacy against E.coli. However, it was reduced in comparison with continuous mode irradiation at 450 mW. The irradiation of P. mirabilis by a pulsed laser at 300 mW output power led to a reduced aPDT efficacy compare to the continuous mode irradiation at the same power. The aPDT efficacy after irradiation at 450 mW by a pulsed laser increased in comparison with irradiation at 300 mW. However, it was lower than after continuous mode irradiation at the same power.

6) Discussion: in relation to what is written in line 260-262, the production of ROS should be determined.

The sentence was rewritten. We do not state that the production of ROS is insufficient, we suppose it to be insufficient. Our next paper will be devoted to determination of ROS and products of lipid peroxidation in bacteria after aPDT.

Reviewer 3 Report

aPDT an important area of research in the age of growing antibiotic resistance. However, the outer membrane of Gram-negative bacteria is often a barrier to the uptake of a photosensitizer. Therefore, mechanisms to overcome the outer membrane for aPDT are an important part of this research.

In the article “Enhancement of the efficacy of photodynamic therapy against uropathogenic Gram-negative bacteria species”, Elagin and coauthors investigated the efficacy of Fotoditazin treatment of four isolated Gram-negative bacteria strains (Escherichia coliKlebsiella pneumoniaePseudomonas aeruginosa and Proteus mirabilis) in presence of the detergent Triton-X-100. Triton X-100 is one of the most popular non-ionic surfactants used for permeabilizing cells.

It is well known that substances weaken the outer membrane (e.g. polyethylenimine, or outer membrane-destabilizing antibiotics like colistin) can improve the efficacy of a photosensitizer against Gram-negative bacteria.

The topic of the study is very important and interesting, however, I have some criticisms in the implementation:

1) Introduction: I must actually confess that I had difficulty finding information about Fotoditazin. As I understood, Fotoditazin is used more for tumors, less against bacteria. Here some more information in the introduction would be desirable - also regarding the application against bacteria. The group has already published an article on this topic (doi:10.3390/photonics8110495). In addition, the photosensitizer used should also be mentioned in the abstract!

2) Chapter 3.1: In Figure 1, I miss data for the statement “The addition of either Tween-80 or Triton X-100 to bacteria without photosensitizer did not cause a significant variation in the number of living cells”. Both the detergent alone and the photosensitizer alone should be listed as controls. It would also be nice to have information on the photosensitizer / detergent concentration in the figure or figure caption (by the way, this applies to all figures).

3) The figures could also be improved a bit. In Figure 1, the percentace numbers overlap with the error bars. What does the aterisk refer to? I guess significance vs. w/o? (this also applies to all figures)

4) Chapter 3.2: From the data in Figure 2 I would conclude that - with the exception of P. mirabilis (maybe due to the expression of special major outer membrane proteins of Proteus?)- the photosensitizer is not better taken up by the bacteria in the presence of Triton X-100 within 15 min.

5) Which I think would make more sense for the message of the study: Incubate the bacteria first with Triton X-200 and then do the treatment with your photosensitizer.

6) Also, uptake kinetics would certainly be a useful addition here.

7) Chapter 3.3./3.4.: In Figure 3 and 4 I miss the comparison to untreated cells irradiated with the laser. It is also known that some Gram-negative bacteria (e.g. E. coli) are affected in their growth by light.

Author Response

Dear Reviewer,

First of all, we would like to thank You for the very helpful comments that allowed us to improve the quality of the manuscript. We hope that these changes now make the manuscript suitable for publication in Photonics. Below, please, find our detailed reply to the comments. Our responses are given in yellow.

1) Introduction: I must actually confess that I had difficulty finding information about Fotoditazin. As I understood, Fotoditazin is used more for tumors, less against bacteria. Here some more information in the introduction would be desirable - also regarding the application against bacteria. The group has already published an article on this topic (doi:10.3390/photonics8110495). In addition, the photosensitizer used should also be mentioned in the abstract!

Fotoditazin is N-dimethylglucamine salt of chlorin e6. The photosensitizer is developed and produced by LLC Veta-Grand in the Russian Federation.  "Fotoditazin" is a brand name, however, some researchers use the name "Photodithazine". There are some published studies devoted to antibacterial PDT mediated by Fotoditazin (Photodithazine). We have improved the introduction by adding the corresponding information.

2) Chapter 3.1: In Figure 1, I miss data for the statement “The addition of either Tween-80 or Triton X-100 to bacteria without photosensitizer did not cause a significant variation in the number of living cells”. Both the detergent alone and the photosensitizer alone should be listed as controls. It would also be nice to have information on the photosensitizer / detergent concentration in the figure or figure caption (by the way, this applies to all figures).

The figure and the figure caption have been improved.

3) The figures could also be improved a bit. In Figure 1, the percentace numbers overlap with the error bars. What does the aterisk refer to? I guess significance vs. w/o? (this also applies to all figures)

The figures have been adapted.

4) Chapter 3.2: From the data in Figure 2 I would conclude that - with the exception of P. mirabilis (maybe due to the expression of special major outer membrane proteins of Proteus?)- the photosensitizer is not better taken up by the bacteria in the presence of Triton X-100 within 15 min.

The effect seen in fig 2 may be due to the expression of efflux pumps by strains of E. coli, K. pneumoniae, and P. aeruginosa. They remove the photosensitizer from bacterial cells, which leads to decreased efficacy of aPDT after washing of the photosensitizer.

5) Which I think would make more sense for the message of the study: Incubate the bacteria first with Triton X-200 and then do the treatment with your photosensitizer.

We also considered this approach. However, in our opinion, it has some limitations. On the one hand, it is the time limitation during lithotripsy. On the other hand, bacteria have quite short doubling time, so pores formed in a bacterium cell during incubation with Triton will further be distributed between daughter cells that will limit photosensitizer uptake.

6) Also, uptake kinetics would certainly be a useful addition here.

The uptake kinetics has been added.

7) Chapter 3.3./3.4.: In Figure 3 and 4 I miss the comparison to untreated cells irradiated with the laser. It is also known that some Gram-negative bacteria (e.g. E. coli) are affected in their growth by light.

The data for laser-irradiated untreated cells are not shown in the figures not to overload them. The needed information is given in the text. The efficacy of the laser treatment alone was less than 10%.

Round 2

Reviewer 2 Report

the authors have answered to the various comments

Author Response

Dear Reviewer,

We would like to thank You again. 

Reviewer 3 Report

The authors have already made significant improvements to the manuscript. I have only minor comments where I think minor changes are needed before it can be published.

1) Figure 1: "e6" is not explained anywhere.

2) Figures 2,4,5: The error bars are cut off at the top.

3) Figures 4,5: I still think the laser irradiation data is a useful addition. I don't expect a large effect from irradiation alone either, but since the irradiation energy is higher than some of the other LLLT experiments (~10 mW), I think this information is useful. The figures are not that crowded and in the worst case you can show the data in the appendix/supplementary.

Author Response

Dear Reviewer,

We would like to thank You again. Below, please, find our detailed reply to the comments. Our responses are given in yellow.

1) Figure 1: "e6" is not explained anywhere.

“e6” is an abbreviation of Fotoditazin. The corresponding information have been added to the figures’ caption.

2) Figures 2,4,5: The error bars are cut off at the top.

The figures have been improved.

3) Figures 4,5: I still think the laser irradiation data is a useful addition. I don't expect a large effect from irradiation alone either, but since the irradiation energy is higher than some of the other LLLT experiments (~10 mW), I think this information is useful. The figures are not that crowded and in the worst case you can show the data in the appendix/supplementary.

The results of treatment of bacteria by laser alone have been included to the subsection 3.3. The respective figure has been added to a supplementary material.